# STAREG: Statistical replicability analysis of high throughput experiments with applications to spatial transcriptomic studies

Yan Li [1,2], Xiang Zhou[3], Rui Chen[4], Xianyang Zhang[5], Hongyuan Cao[6]*

1 School of Computer Science and Technology, Changchun University of Science and Technology, Changchun, Jilin, China, 2 School of Mathematics, Jilin University, Changchun, Jilin, China, 3 Department of Biostatistics, University of Michigan, Ann Arbor, Michigan, United States of America, 4 Department of Molecular and Human Genetics, Baylor College of Medicine, Houston, Texas, United States of America, 5 Department of Statistics, Texas A&M University, College Station, Texas, United States of America, 6 Department of Statistics, Florida State University, Tallahassee, Florida, United States of America

* hongyuancao@gmail.com

**Data Availability Statement:** All data are publicly available and can be acquired from corresponding websites. The mouse olfactory bulb data can be downloaded from the Spatial Research Website at

## Abstract

Replicable signals from different yet conceptually related studies provide stronger scientific evidence and more powerful inference. We introduce STAREG, a statistical method for replicability analysis of high throughput experiments, and apply it to analyze spatial transcriptomic studies. STAREG uses summary statistics from multiple studies of high throughput experiments and models the the joint distribution of *p*-values accounting for the heterogeneity of different studies. It effectively controls the false discovery rate (FDR) and has higher power by information borrowing. Moreover, it provides different rankings of important genes. With the EM algorithm in combination with pool-adjacent-violator-algorithm (PAVA), STAREG is scalable to datasets with millions of genes without any tuning parameters. Analyzing two pairs of spatially resolved transcriptomic datasets, we are able to make biological discoveries that otherwise cannot be obtained by using existing methods.

## Author summary

Irreplicable research wastes time, money, and/or resources. Approximately $28 billion is estimated to be spent on preclinical research that cannot be replicated every year in the United States alone. Possible causes of irreplicable research may include experimental design, laboratory practices, and data analysis. We focus on data analysis. The past two decades have witnessed the expansion and increased availability of genomic data from high-throughput experiments. Due to privacy concerns or logistic reasons, raw data can be difficult to access but summary data such as *p*-values are readily available. We introduce STAREG, which jointly analyzes *p*-values from multiple genomic datasets that target the same scientific question with different populations or different technologies. This allows us to have more convincing and robust findings. STAREG is computationally scalable with solid statistical analysis. Moreover, it is versatile, platform-independent, and only requires *p*-values as input. By analyzing data sets from spatially resolved

https://www.spatialresearch.org/resources-published-datasets/doi-10-1126science-aaf2403 (files "MOB Replicate 1" and "MOB Replicate 8"). The Slide-seq data and Slide-seqV2 data from the mouse cerebellum are available at the Broad Institute's single-cell repository (https://singlecell.broadinstitute.org/single_cell/) under IDs SCP354 (file "Puck_180819_12") and SCP948, respectively. R package STAREG is available at https://cran.r-project.org/package=STAREG. The R scripts and underlying raw data to reproduce our analyses are freely available at GitHub at https://github.com/hongyuan-cao/STAREG-Analysis.

**Funding:** The author(s) received no specific funding for this work.

**Competing interests:** The authors have declared that no competing interests exist.

transcriptomic studies, we make biological discoveries that otherwise cannot be obtained with existing methods.

## Introduction

Replicability is a cornerstone of modern scientific research. Consistent results from different studies with different data provide more robust scientific evidence. In addition, pooling inferences made under different yet conceptually related conditions enables researchers to gain statistical power. Replicability has attracted considerable attention, scrutiny, and debate in the popular press and scientific community. An analysis of past studies indicates that the cumulative prevalence of irreplaceable pre-clinical research exceeds 50% [1]. Approximately $28 billion is spent annually on preclinical research that is not replicable in the United States alone [2]. Possible causes of irreplicability may range from study design, biological reagents and reference materials, laboratory protocols, and data analysis, among others. We focus on data analysis and study conceptual replicability where consistent results are obtained using different processes and populations that target the same scientific question with genomic data from high throughput experiments.

With hundreds of thousands of genes from high throughput experiments of multiple studies, an acute problem is multiple comparisons. To identify statistically replicable significant associations, an *ad hoc* approach is first to compute *p*-values from each study, then apply a multiple comparison procedure, such as the [3] (BH) false discovery rate (FDR) control procedure, separately for each study, and finally declare replicable genes as the intersection of significant genes from different studies. This approach does not control the FDR and has low power, as it does not borrow information from different studies. As a conservative alternative, researchers may use the maximum of *p*-values from different studies for each gene and implement a multiple testing procedure to claim replicability [4]. This can lose substantial power, as demonstrated in our simulation studies and data analysis. An improvement in using the maximum of *p*-values was recently developed in [5] and [6]. For high-throughput experiments, [7] developed the irreproducible discovery rate to measure replicability, and [8] proposed a non-parametric approach for replicability assessment. Both methods impose the strong modeling assumption that genes from two studies are either both significant or non-significant. A Bayesian method incorporating heterogeneity of different studies was proposed in [9], where a tuning parameter has to be selected. In addition, [10] used a cross-fitting idea to reduce the number of multiple comparisons and borrow information from two studies. [11] proposed an empirical Bayesian method where the density function of *p*-values under non-null was modeled parameterically. Recently, [12] proposed criteria for replicability assessment, and a survey on this topic can be found in [13].

In this paper, we propose a robust and powerful approach for replicability analysis of genomic data from high throughput experiments. We apply it to the spatial transcriptomic analysis of replicable expressed genes (STAREG) from two spatially resolved transcriptomic studies. Spatially resolved transcriptomic (SRT) studies profile high throughput gene expression while preserving spatial location information of cells in tissues or cell cultures [14–17]. This additional dimension of spatial information brings new perspectives on the cellular transcriptome, allowing researchers to uncover complex cellular and sub-cellular architecture in heterogeneous tissues, which provides crucial insights into complex biological processes [16]. In SRT studies, genes that show spatial expression variations across spatial locations on a tissue section are known as spatially variable genes (SVGs). Detecting SVGs is an important first step in

characterizing the spatial transcriptomic landscapes of complex tissues [18]. Various methods have been developed to detect SVGs such as SpatialDE [19], SPARK [20], Giotto [21], and SPARK-X [22], among others. A comprehensive evaluation of different SVG detection methods can be found in [23]. We use spatial transcriptomic data from two independent yet conceptually highly related samples to illustrate STAREG and remark that STAREG is general and not limited to spatial transcriptomic studies. For instance, when analyzing olfactory bulb data from two mice, a true SVG should exhibit spatial patterns in both mice though the spatial patterns and tissue sections can be different.

Based on the hidden state of whether a gene is an SVG or not, we extend the two-group model to a four-group model [24–26] to account for the heterogeneity of different studies. A gene is a replicable SVG if it is significant in both studies and the null hypothesis is composite, consisting of three different states. We do not need the two SRT studies to have the same alignment, as rotations or normalizations may distort crucial biological information. In addition, we estimate the $p$-value density functions under the non-null non-parametrically and differently for different studies to account for the heterogeneity of signal strengths and/or sample sizes of different studies. We use the local false discovery rate (Lfdr), the posterior probability of being null given the data, as the test statistic [24]. Lfdr combines the information in null and non-null hypotheses and thus provides different rankings of importance compared to $p$-value-based methods, such as the BH procedure, which is obtained only under the null hypothesis. Moreover, our method enumerates different states in the composite null and has good power. A step-up procedure is used to obtain an asymptotic FDR control [27]. By borrowing information across genes and different studies, STAREG is more powerful at detecting replicable SVGs while controlling the FDR. By combining EM algorithm and pool-adjacent-violator-algorithm (PAVA), STAREG is scalable to datasets with tens of thousands of genes measured on tens of thousands of spatial spots without any tuning parameters [28–30]. We use STAREG to analyze two pairs of SRT datasets to detect replicable SVGs. The first pair are data from the mouse olfactory bulb (MOB) measured in two mice with ST technology [14]. The second pair are mouse cerebellum (MC) data measured with Slide-seq technology [15] and Slide-seqV2 technology [17]. We show that STAREG uncovers important replicable biological discoveries that cannot be made otherwise by existing methods.

## Results

### Method overview

Suppose we have two datasets obtained by measuring the spatial expression patterns of $m$ genes under distinct cellular environments with different technologies. We describe the workflow of STAREG for replicability analysis of SVG detection from two SRT studies. A schematic of STAREG is shown in Fig 1. For each SRT dataset, we test $m$ hypotheses simultaneously, where the null hypothesis for $i$th gene states that it is not a spatially variable gene, and the non-null hypothesis states that it is a spatially variable gene. We require $p$-values to follow standard uniform distribution under the null for both studies. Any statistical methods that produce well-calibrated $p$-values can be used in this step.

STAREG uses paired $p$-values $(p_{1i}, p_{2i})$, $i = 1, \ldots, m$ as input to the replicability analysis. Let $\theta_{ji}$ denote the hidden state of $i$th hypothesis in study $j$ ($j = 1, 2$), where $\theta_{ji} = 1$ indicates $i$th gene is significant in study $j$, and $\theta_{ji} = 0$ otherwise. Write the joint hidden states across two studies as $s_i \in \{(0, 0), (0, 1), (1, 0), (1, 1)\}$ with prior probabilities $P(s_i = (k, l)) = \xi_{kl}$, where $k, l = 0, 1$ and $\sum_{k,l} \xi_{kl} = 1$. Our $i$th null hypothesis for replicability of SVG in two studies is $H_{i0} : s_i \in \{(0, 0), (0, 1), (1, 0)\}$, $i = 1, \ldots, m$. We use a four-group model for the joint distribution of $(p_{i1}, p_{i2})$, $i = 1, \ldots, m$, where the distribution of $p$-values under the null is assumed to be standard

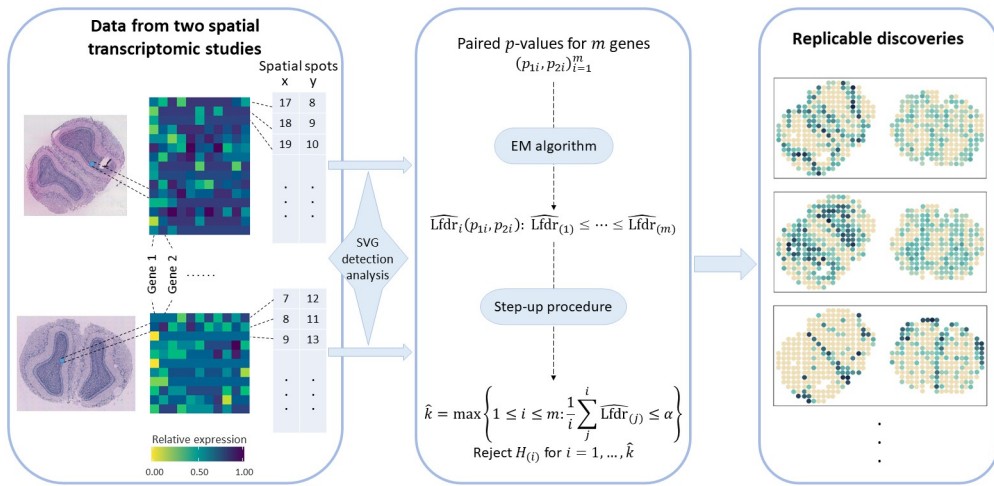

**Fig 1. Schematic of STAREG for identifying replicable SVGs from two SRT studies.**

uniform for both studies and $p$-value distributions under the non-null are estimated non-parametrically and can be different in two SRT studies to accommodate heterogeneity. For $i$th gene, we use Lfdr as our test statistic, which is defined as the posterior probability of $H_{i0}$ given $(p_{i1}, p_{i2})$, $i = 1, \ldots, m$. We use the EM algorithm in combination with PAVA to estimate unknown parameters $(\xi_{00}, \xi_{01}, \xi_{10}, \xi_{11})$ and $p$-value distributions under the non-null for study 1 and study 2 non-parametrically. Denote estimated Lfdr as $\widehat{\text{Lfdr}}_i$, $i = 1, \ldots, m$. Small Lfdr indicates strong evidence against the null. The rejection region can be written as $\delta_i = \{\widehat{\text{Lfdr}}_i \leq \lambda\}$. We implement a step-up procedure based on $\widehat{\text{Lfdr}}_i$, $i = 1, \ldots, m$ to identify replicable SVGs from the two SRT studies [27]. More details of STAREG can be found in the Methods Section and Section A in S1 Appendix.

## Simulation studies

We performed simulation studies to evaluate FDR control and power of STAREG. FDR is defined as the expectation of the number of false rejections over the total number of rejections. Power refers to the true positive rate, which is the expectation of the number of replicable findings over the total number of non-null hypotheses. In each simulation, we used pre-specified $\xi_{00}, \xi_{01}, \xi_{10}$ and $\xi_{11}$ to generate hidden states $(\theta_{1i}, \theta_{2i})$, $i = 1, \ldots, m$, where $P(\theta_1 = k, \theta_2 = l) = \xi_{kl}$, $k, l = 0, 1$, for $m$ genes from multinomial distribution.

**Simulations based on normal distributions.** First, we performed numerical studies based on normal distributions and $z$-statistics to evaluate FDR control and power of STAREG. Denote $N(\mu, \sigma^2)$ a normal distribution with mean $\mu$ and variance $\sigma^2$. For study $j$, we independently generated summary statistics from $X_{ji} \sim N(0, \sigma_j^2)$ if $\theta_{ji} = 0$, and from $X_{ji} \sim N(\mu_j, \sigma_j^2)$ if $\theta_{ji} = 1$, where $\mu_j > 0$. One-sided $p$-value for each gene was calculated by $p_{ji} = 1 - \Phi(Z_{ji})$, $j = 1, 2$; $i = 1, \ldots, m$, where $Z_{ji} = X_{ji}/\sigma_j$ denotes the $Z$-statistic for the $i$th gene in study $j$ and $\Phi(\cdot)$ is the cumulative distribution function of standard normal distribution $N(0, 1)$.

Set $m = 10, 000$, $\xi_{11} = 0.05$ and let $\xi_{01} = \xi_{10}$ take values from 0.05 to 0.2. Corresponding $\xi_{00}$ can be calculated by $\xi_{00} = 1 - \xi_{01} - \xi_{10} - \xi_{11}$. At a target FDR level of 0.05, FDR and power of different methods were calculated from 100 runs in each setting. Fig 2A and 2B show the empirical FDR and power over a range of values for $\xi_{01}$ under different non-null settings,

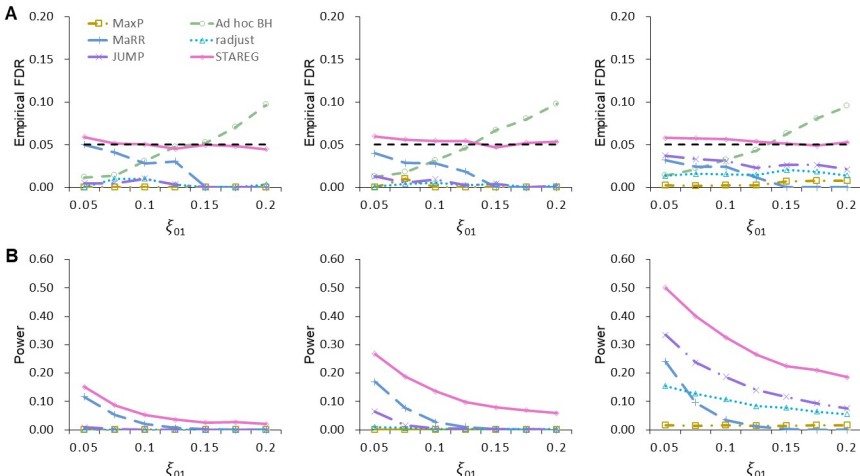

**Fig 2. Performance comparisons of different methods in simulation studies based on normal distributions with different $\xi_{01}$.** Left: $\mu_1 = \mu_2 = 2$, $\sigma_1 = \sigma_2 = 1$. Middle: $\mu_1 = 2$, $\mu_2 = 2.5$, $\sigma_1 = \sigma_2 = 1$. Right: $\mu_1 = \mu_2 = 2$, $\sigma_1 = 1$, $\sigma_2 = 0.5$. All simulations were conducted under the setting of $m = 10,000$, $\xi_{11} = 0.05$ and $\xi_{01} = \xi_{10}$. The target FDR level is 0.05 represented by horizontal dashed lines in (**A**), and the results were calculated over 100 runs. We do not present the power of *ad hoc* BH in (**B**) as it fails to control FDR.

respectively. We see that *ad hoc* BH failed to control FDR when the inconsistency between two studies is relatively large (e.g., $\xi_{01} = \xi_{10} > 0.15$). MaxP was overly conservative in all settings. MaRR, radjust and JUMP showed decent power while maintaining FDR control. STAREG properly controlled FDR in all settings and had the highest power at the same nominal FDR level. At target FDR levels from 0 to 1, we plot the power of different methods with the corresponding empirical FDR under the setting of $\xi_{00} = 0.9$, $\xi_{01} = \xi_{10} = 0.025$ and $\xi_{11} = 0.05$ (Fig 3A). Moreover, Fig 3B presents the receiver operator characteristic (ROC) curves (with false positive rate as the horizontal axis and true positive rate as the vertical axis) and corresponding area under curve (AUC) values of different methods from one run. The false positive rate is calculated as the number of false discoveries divided by the total number of true nulls and the true positive rate is equivalent to the power. We observe that STAREG showed higher power than competing methods across all settings.

We evaluated the computational time of STAREG in simulation studies with $\xi_{00} = 0.9$, $\xi_{01} = \xi_{10} = 0.025$, $\xi_{11} = 0.05$, $\mu_1 = \mu_2 = 2$ and $\sigma_1 = \sigma_2 = 1$. Table 1 summarizes the computational time of different methods for different numbers of genes. All methods are implemented in R, in which we use *Rcpp* to speed up the computation. Computations were carried out in an i7-9750H 2.6GHz CPU with 64.0 GB RAM laptop. We observe that all methods are quick to compute. STAREG takes less time than MaRR and longer time than the other methods, though such differences can be ignored in practical data analysis.

**Realistic simulations based on SRTsim.** We next performed realistic simulations by generating spatial transcriptomic data from SRTsim [31]. SRTsim maintains various expression characteristics of SRT data and preserves spatial patterns. We separately generated gene expression count data for two studies using SRTsim based on simulated $(\theta_{1i}, \theta_{2i})$, $i = 1, \ldots, m$. We generated 384 and 486 spatial location coordinates based on circular tissue shapes by random sampling for the two studies, respectively. Gene count data were generated by the zero-inflated Poisson model embedded in SRTsim. The dispersion, zero-proportion, and mean parameters were set to 0.2, 0.01, and 2, respectively. The spatial spots for each study were divided into two spatial domains. For non-SVGs, the fold change is set to 1 across all spots; for

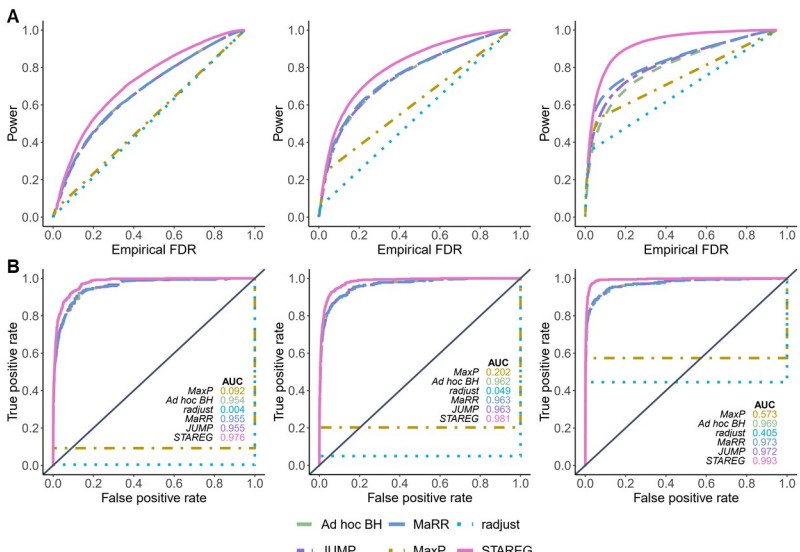

**Fig 3. Performance comparisons of different methods in simulation studies based on normal distributions with different levels of nominal FDR.** Left: $\mu_1 = \mu_2 = 2$, $\sigma_1 = \sigma_2 = 1$. Middle: $\mu_1 = 2$, $\mu_2 = 2.5$, $\sigma_1 = \sigma_2 = 1$. Right: $\mu_1 = \mu_2 = 2$, $\sigma_1 = 1$, $\sigma_2 = 0.5$. All simulations were performed under the setting of $m = 10,000$, $\xi_{00} = 0.9$, $\xi_{01} = \xi_{10} = 0.025$ and $\xi_{11} = 0.05$. The Receiver Operating Characteristic (ROC) curves and corresponding Area Under Curve (AUC) values in (**B**) were obtained from one replication. The diagonal dashed line with slope 1 is used as a reference.

SVGs, the fold change is set to 1 in the red domain and 3 in the blue domain as in Fig 4. After obtaining the gene expression count matrices for two studies, we applied SPARK-X [22] to produce paired *p*-values as inputs of the replicability analysis.

We set $m = 10,000$, $\xi_{11} = 0.05$ and $\xi_{01} = \xi_{10}$ across all simulations, such that with a specified $\xi_{00}$, we have $\xi_{01} = \xi_{10} = (1 - \xi_{00} - \xi_{11})/2$. We varied $\xi_{00} \in \{0.65, 0.75, 0.85\}$ to evaluate the performance of different methods. For each setting, the empirical FDR and power of different methods are calculated based on 100 simulated datasets. Fig 4 shows the empirical FDR and power with nominal FDR levels ranging from 0.01 to 0.2. We observe that *ad hoc* BH fails to control the FDR when the two studies are relatively heterogeneous (e.g., $\xi_{00} = 0.65$), whereas MaRR was overly conservative. MaxP was conservative across all settings. JUMP and radjust showed decent power while maintaining FDR control. STAREG properly controlled FDR across all settings and had the highest power at the same nominal FDR levels.

Simulation studies evaluating the robustness of our method to dependence among genes and one additional realistic simulation based on parameters inferred from MOB data [14] can be found in the Section C in S1 Appendix.

**Table 1. Computational time (in seconds) for replicability analysis in simulation studies based on normal distributions with different numbers of genes.**

| Method \ # of genes | 5,000 | 10,000 | 20,000 | 5,0000 | 100,000 |
|---|---|---|---|---|---|
| *Ad hoc* BH | 0.0040 | 0.0052 | 0.0056 | 0.0100 | 0.0160 |
| MaxP | 0.0104 | 0.0181 | 0.0367 | 0.0921 | 0.1577 |
| JUMP | 0.0183 | 0.0474 | 0.0515 | 0.1332 | 0.3089 |
| MaRR | 0.6931 | 3.3189 | 8.5864 | 47.077 | 173.56 |
| radjust | 0.0103 | 0.0098 | 0.0107 | 0.0154 | 0.0258 |
| STAREG | 0.0402 | 0.0584 | 0.0705 | 0.2569 | 0.4209 |

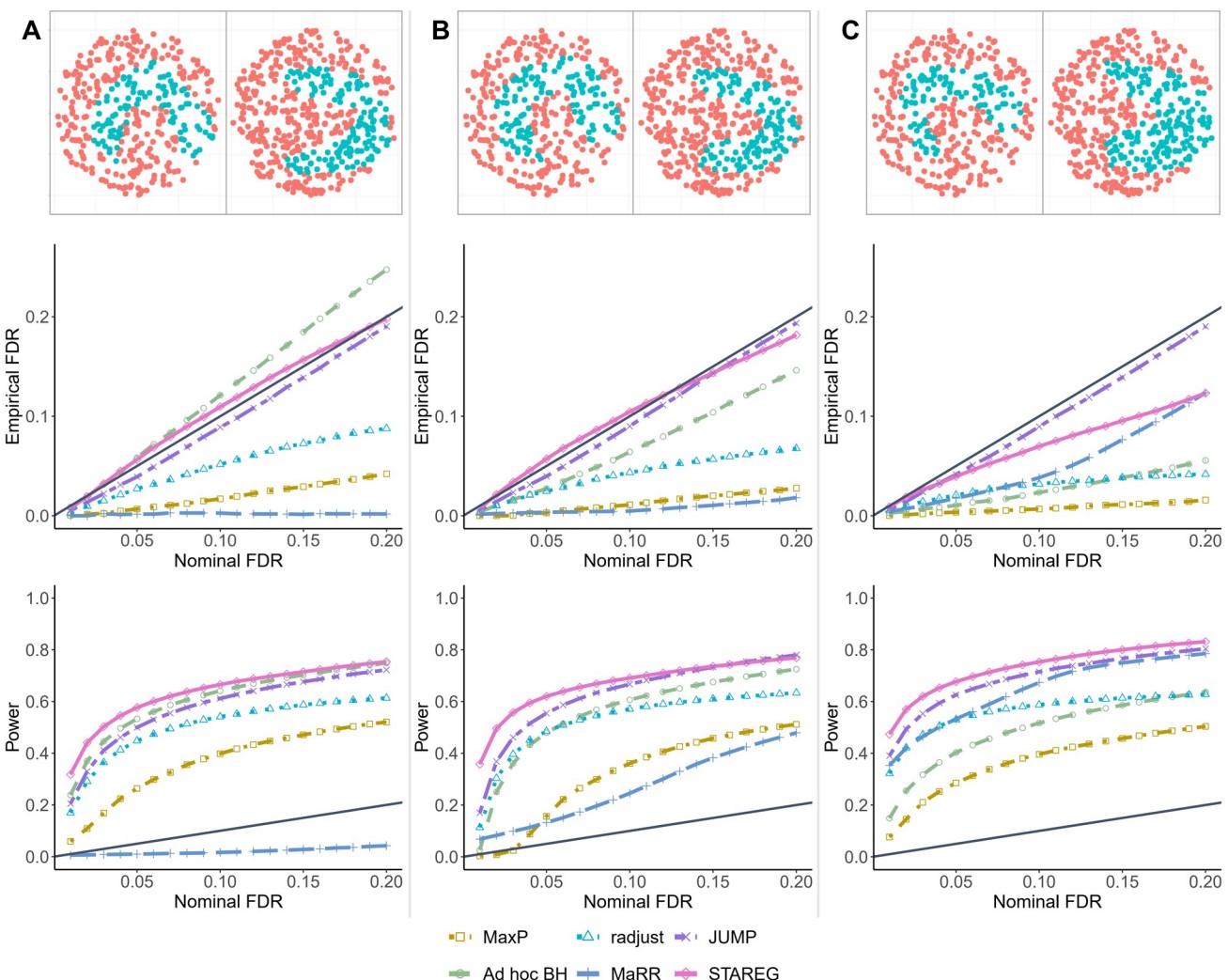

**Fig 4. Empirical FDR and power of different methods in realistic simulations based on SRTsim [31].** Each column corresponds to a different value of $\xi_{00}$ (**A**: $\xi_{00} = 0.65$; **B**: $\xi_{00} = 0.75$; **C**: $\xi_{00} = 0.85$.) Simulations were conducted under the setting of $m = 10,000$, $\xi_{11} = 0.05$ and $\xi_{01} = \xi_{10}$. The nominal FDR level ranges from 0.01 to 0.2, and the results were calculated over 100 replications.

## Data analysis

**Analyzing mouse olfactory bulb data.** The datasets contain two replicates of mouse olfactory bulb data [14] published on the Spatial Research Website (https://www.spatialresearch.org/). The Replicate 1 data (file "MOB Replicate 1") consist of 16, 573 genes on 265 spots, and the Replicate 8 data (file "MOB Replicate 8") contain 15, 288 genes on 234 spots. For each dataset, we filtered out genes that are expressed in less than 10% of the spatial locations and selected spatial locations with at least ten total read counts, resulting in 10, 373 genes on 265 spots in Replicate 1 dataset and 9, 671 genes on 232 spots in Replicate 8 dataset. We applied SPARK [20] separately on the two datasets to get two sequences of *p*-values. We next took 9, 329 pairs of *p*-values of common genes in both studies as input for replicability analysis. We also conducted analysis of the MOB data based on *p*-values from SPARK-X [22] in Section D.1 in S1 Appendix as a comparison.

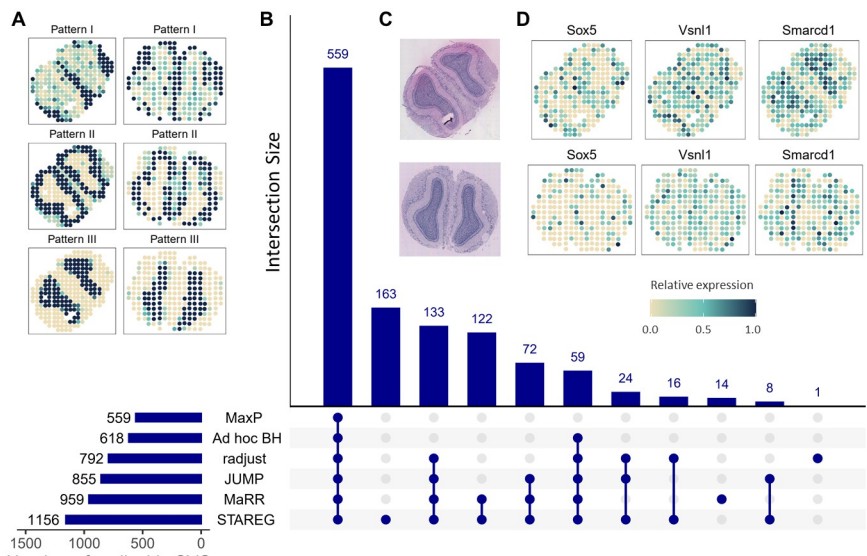

**Fig 5. MOB data analysis results at FDR level 0.05.** (**A**) Three spatial patterns summarized based on 1, 156 replicable SVGs identified by STAREG (left: ST Replicate 1 study; right: ST Replicate 8 study). (**B**) The number of replicable SVGs identified by different methods. (**C**) HE images of the MOB Replicate 1 section (top) and MOB Replicate 8 section (bottom). (**D**) Spatial patterns of three representative replicable SVGs only identified by STAREG based on the Replicate 1 study (top) and the Replicate 8 study (bottom).

The results of different methods at FDR level 0.05 are summarized in Fig 5B. STAREG detected 1, 156 replicable SVGs, including 163 replicable SVGs that were not identified by any other methods. MaxP identified 559 replicable SVGs, which were also detected by all other methods. The *ad hoc* BH identified 618 replicable SVGs, radjust identified 792 replicable SVGs (1 of which was not identified by STAREG), JUMP identified 855 replicable SVGs, and MaRR identified 959 replicable SVGs (14 of which were not identified by STAREG). We plot the non-parametric estimates of *p*-value density functions under non-null for two studies in S1A Fig. To assess the quality of the 1, 156 replicable SVGs identified by STAREG, we clustered these genes into three groups with distinct spatial expression patterns using R package *amap* v0.8–18. As shown in Fig 5A, the three distinct spatial patterns in the two studies (top: Replicate 1 study; bottom: Replicate 8 study) are consistent and can be matched to three main layers in MOB: Pattern I corresponds to the glomerular cell layer, Pattern II corresponds to the mitral cell layer and Pattern III corresponds to the granular cell layer. Hematoxylin and eosin (HE) staining images representing the mouse olfactory bulb sections are presented in Fig 5C (top: Replicate 1 study; bottom: Replicate 8 study). Spatial expression patterns of *Sox5*, *Vsnl1* and *Smarcd1*, three representative genes only detected by STAREG, are presented in Fig 5D. We calculated Moran's *I* statistic [32] to quantify spatial autocorrelations of the replicable SVGs only identified by STAREG. As in Fig 6A, the 163 SVGs only identified by STAREG have larger Moran's *I* than that of the 8, 173 genes not detected by STAREG. S2A Fig presents three spatial patterns summarized based on the 163 replicable SVGs only identified by STAREG. The three distinct spatial patterns can be matched to three main layers in MOB. Spatial expression patterns of 16 randomly selected SVGs only identified by STAREG are listed in S2B and S2C Fig as additional evidence.

We summarized two lists of reference genes related to MOB from previously published literature to validate replicable SVGs identified by different methods. The validation rate is

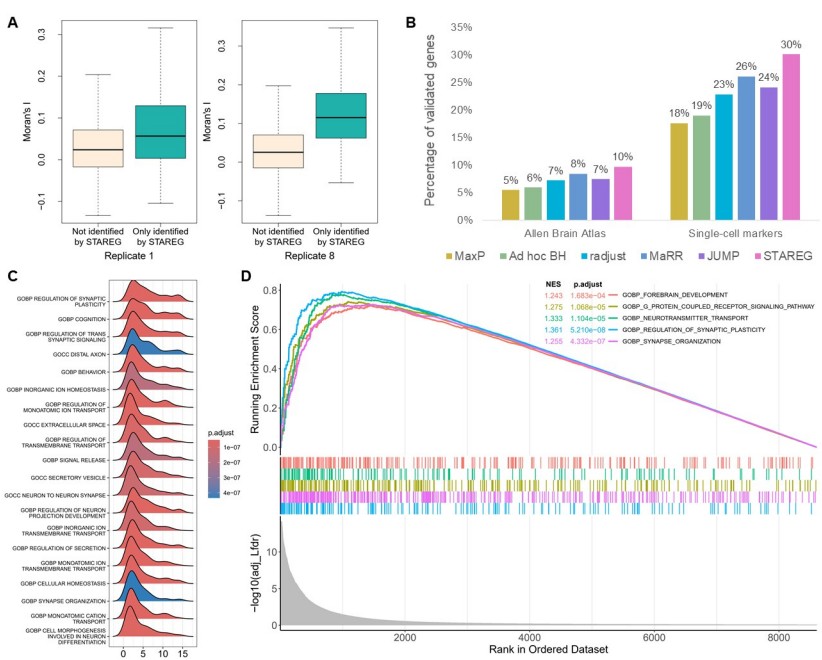

**Fig 6. Validation results of the MOB data analysis. (A)** Moran's *I* statistic of the 163 replicable SVGs only identified by STAREG compared to genes not identified by STAREG based on ST Replicate 1 study (left) and ST Replicate 8 study (right). **(B)** The percentage of validated genes in the corresponding reference gene list based on replicable SVGs identified by different methods. **(C)** The ridge plot shows the distribution of genes in the top-20 gene sets enriched by STAREG. **(D)** The running enrichment scores of five representative gene sets only enriched by STAREG at FDR cutoff 0.05.

calculated as the number of validated genes within each detected replicable SVG set divided by the total number of genes in corresponding reference gene list. The first reference gene set from the Allen Brain Atlas [33] (https://maayanlab.cloud/Harmonizome/) contains 3, 485 genes differentially expressed in three main layers in MOB relative to other tissues. As can be seen in the left part of Fig 6B, STAREG has the highest validation rate (10%), followed by MaRR (8%), JUMP (7%), radjust (7%), *ad hoc* BH (6%) and MaxP (5%). Second, we obtained a list of 2030 cell-type marker genes identified in a recent single-cell RNA sequencing study in MOB [34]. As in the right part of Fig 6B, STAREG has the highest validation rate of 30% in the Single-cell marker gene list, followed by MaRR (26%), JUMP (24%), radjust (23%), *ad hoc* BH (19%) and MaxP (18%). Furthermore, we examined the top-2, 000 gene set based on the maximum of *p*-values and the Lfdr estimated by STAREG. We observe that the top-2, 000 gene set based on Lfdr has higher validation rate in both reference gene lists (S3A Fig).

Finally, we performed gene set enrichment analysis (GSEA) for the gene ontology (GO) gene sets [35] with the R package *fgsea* [36] to gain additional biological insights. STAREG ranked genes based on the average of the order statistics of Lfdr. Specifically, let $\widehat{\text{Lfdr}}_{(1)} \leq \ldots \leq \widehat{\text{Lfdr}}_{(m)}$ be the order statistics of $\{\widehat{\text{Lfdr}}_i\}_{i=1}^m$, then genes were ranked based on $\frac{1}{i}\sum_{j=1}^{i}\widehat{\text{Lfdr}}_{(j)}, i = 1, \ldots, m$. In comparison, a common strategy for ranking genes is using *p*-values. Specifically, suppose $p_i$ is the maximum of *p*-values from the two studies for the *i*th gene, and $p_{(1)} \leq \ldots \leq p_{(m)}$ denote the order statistics. Genes were ranked based on $mp_{(i)}/i$, $i = 1, \ldots, m$. Using Lfdr based ranking, 781 GO terms were enriched at FDR cutoff 0.05 (Fig 6C and 6D), including 4 of the 5 GO terms identified by *p*-value based ranking. Many of the

additional identified GO terms are related to synapse assembly, dendrite morphogenesis, neuron differentiation, and regulation of G-protein, which play critical roles in olfactory bulb organization and olfactory signal transduction [37]. These additional enrichments in GO terms demonstrate the biological significance of findings only detected by STAREG. We present GO enrichment analysis of the 163 replicable SVGs only identified by STAREG in S3B Fig.

**Analyzing mouse cerebellum data.** Slide-seq is a technology that provides scalable methods for obtaining spatially resolved transcrimptotic data at resolutions comparable to single cells [15]. Slide-seqV2 is built on Slide-seq, combining improvements in library generation, bead synthesis, and array indexing to achieve a ten-fold higher RNA capture efficiency than Slide-seq [17]. We obtained two datasets of mouse cerebellum measured with Slide-seq and Slide-seqV2 from Broad Institute's single-cell repository (https://singlecell.broadinstitute.org/single_cell), with IDs SCP354 and SCP948, respectively. For the Slide-seq data, we used the dataset summarized in the file "Puck_180819_12", which contains 19, 782 genes measured on 32, 701 beads. The Slide-seqV2 data contains 23, 096 genes on 39, 496 beads. For the Slide-seq dataset, we first filtered out beads that were not assigned clusters in the original study [15]. For the Slide-seqV2 dataset, we cropped regions of interest by filtering out beads with UMIs less than 100 following [38]. Mitochondrial genes and genes that were not expressed on any location were filtered out from the two datasets, and beads with zero total expression counts were removed, resulting in 18, 082 genes on 28, 352 beads in the Slide-seq dataset and 20, 117 genes on 11, 626 beads in the Slide-seqV2 dataset. Due to the computational complexity of these datasets, we applied SPARK-X [22] to analyze them separately, resulting in two sequences of $p$-values from corresponding studies. By intersecting genes in these two studies, we performed replicability analysis of SVG detection on 16, 873 pairs of $p$-values.

As shown in Fig 7B, at FDR cutoff of 0.05, MaxP identified 286 replicable SVGs, all of which were identified by all other methods. STAREG identified 875 replicable SVGs, 341 of which otherwise cannot be detected by any other competing methods. The *ad hoc* BH

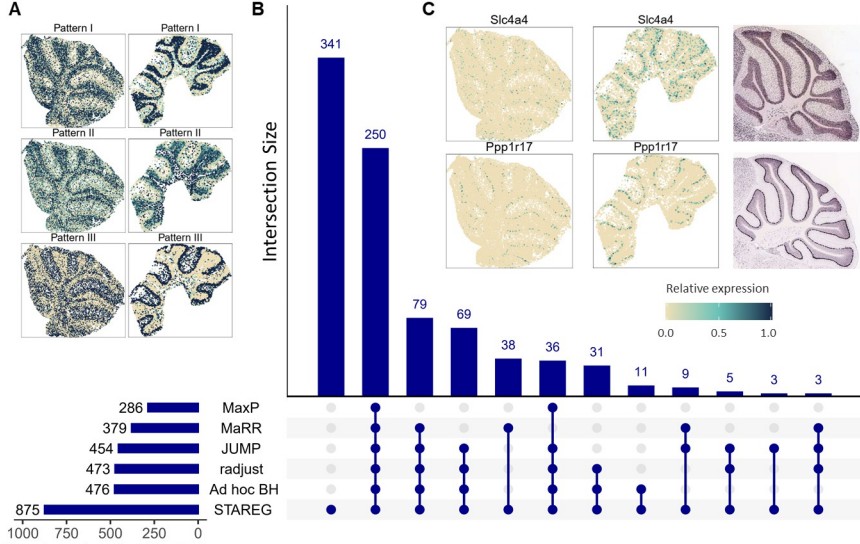

**Fig 7. MC data analysis results at FDR level 0.05.** (**A**) Three spatial patterns summarized based on 875 replicable SVGs identified by STAREG (left: Slide-seq study; right: Slide-seqV2 study). (**B**) The number of replicable SVGs identified by different methods. (**C**) Spatial expression patterns of two representative replicable SVGs only identified by STAREG based on the Slide-seq study (left) and the Slide-seqV2 study (right). The *in situ* hybridization images of corresponding genes obtained from the Allen Brain Atlas are displayed on the right as references.

identified 476 replicable SVGs, radjust identified 473 replicable SVGs, JUMP identified 454 replicable SVGs, and MaRR identified 379 replicable SVGs. We plot the non-parametric estimates of *p*-value density functions under non-null for two studies in S1B Fig. Based on these two datasets, we clustered the 875 replicable SVGs identified by STAREG into three groups and summarized distinct spatial expression patterns (Fig 7A). We observe consistent spatial patterns in the two studies, with Pattern I corresponding to the spatial distribution of the granular cell layer, Pattern III corresponding to the purkinje cell layer, and Pattern II corresponding to other cell layers. STAREG uniquely identified several well-known cell type marker genes in MC, such as *Ppp1r17* [39], *Gabra1* [40], *Edil3* [41], *Gdf10* [42], *Ptprk* and *Nxph1* (Allen brain atlas). We list the spatial expression pattern and corresponding *in situ* hybridization images of two replicable SVGs only identified by STAREG, *Slc4a4* and *Ppp1r17*, based on the Slide-seq data and Slide-seqV2 data as examples of the granular layer and purkinje layer in mouse cerebellum (Fig 7C). S4A Fig presents three spatial patterns summarized based on the 341 replicable SVGs only identified by STAREG. The spatial expression patterns of 10 randomly selected SVGs only identified by STAREG are listed in S4B Fig as additional evidence. Spatial autocorrelations of the 341 replicable SVGs only detected by STAREG compared to the 15, 998 genes not identified by STAREG were further demonstrated by Moran's *I* statistics [32] in Fig 8A).

We validated the quality of replicable SVGs identified by different methods using two lists of genes related to MC that were published in previous literature. The first validation set consists of 2, 431 genes related to MC from the BioGPS mouse cell type and tissue gene expression profiles dataset [43]. As in the left part of Fig 8B, 10% of genes detected by STAREG were validated. The validation rates of the other methods, in descending order, were 6% (*ad hoc* BH

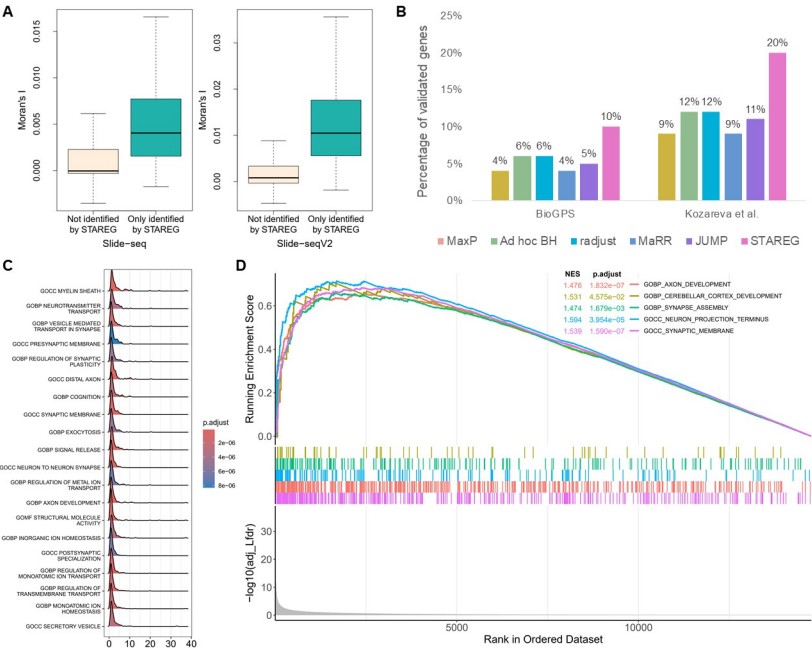

**Fig 8. Validation results of the MC data.** (**A**) Moran's *I* statistic of the 341 replicable SVGs only identified by STAREG compared to genes not identified by STAREG based on Slide-seq study (left) and Slide-seqV2 study (right). (**B**) The percentage of validated genes in the corresponding reference gene lists [43, 44] based on replicable SVGs identified by different methods. (**C**) The ridge plot shows the distribution of genes in the top-20 gene sets enriched by STAREG. (**D**) The running enrichment scores of five representative gene sets only enriched by STAREG at FDR cutoff 0.05.

and radjust), 5% (JUMP), and 4% (MaxP and MaRR). Second, we obtained a list of differentially expressed genes across all cell clusters in mouse cerebellar cortex from [44]. Let log FC denote logarithmic fold changes. By filtering out genes with |log FC| < 1.5, we used a final set of 995 marker genes that are differentially expressed genes across all cell clusters in mouse cerebellar cortex for the validation. Fig 8B (right) shows the validation results. STAREG has the highest validation rate of 20%, followed by *ad hoc* BH (12%), radjust (12%), JUMP (11%), MaRR (11%) and MaxP (11%). We also checked the top-2, 000 gene set based on the maximum of $p$-values and the Lfdr estimated by STAREG. As in S5A Fig, the top-2, 000 gene set based on Lfdr has higher validation rate in both reference gene lists.

Finally, we performed GSEA for the GO gene sets with the R package *fgsea* to examine the additional biological findings. At FDR cutoff 0.05, 448 GO terms were enriched by using Lfdr based ranking (Fig 8C and 8D), while 6 GO terms were identified by $p$-value based ranking. The GO terms enriched by adjusted Lfdr reveal critical roles of synaptic signaling, axon development, neuron projection, and cerebellar cortex development for the cerebellum structure and functions. These additional biologically relevant findings make STAREG desirable for replicability analysis of SVG detection. This can be further illustrated by the GO enrichment analysis of 341 replicable SVGs only identified by STAREG (S5B Fig).

## Discussion

In this paper, we present a powerful new method for the detection of replicable signals across two studies, and illustrate its application to replicable SVG detection. By borrowing information across genes and different studies, STAREG has higher statistical power while maintaining asymptotic FDR control. We conducted extensive simulation studies to demonstrate FDR control and power gain of STAREG over competing methods. Analysis results of data from different species, regions, and tissues generated by different spatially resolved transcriptomic technologies demonstrate the favorable performance of STAREG. Important biological findings are revealed by STAREG, which otherwise cannot be obtained by using existing methods.

STAREG is versatile, platform-independent, and only needs $p$-values as input. We require $p$-values to be uniformly distributed under the null. Such a requirement is needed in other methods. In the analysis of spatially resolved transcriptomic data, we do not require two tissue sections to have the same spatial expression patterns, making STAREG applicable for data with different sizes, resolutions, and alignments obtained from different spatially resolved transcriptomic technologies. In practice, using alignments, rotations, and normalizations may distort important biological information contained in the data. Moreover, although we focus on the replicability analysis of SVG detection from two studies, STAREG can be easily extended to other modalities, such as scRNA-seq, ATAC-seq, and CITE-seq, among others. We provide one scRNA-seq data analysis in the Section D.2 in S1 Appendix.

A limitation is that the current version of STAREG only considers the replicability analysis of two SRT studies. In theory, it is straightforward to extend the current method to more than two studies since we only need to add additional components to the mixture model. The derivation can be found in Section E.1 and E.2 in S1 Appendix. In practice, this can be computationally expensive if the number of studies is large, as the computational time scales exponentially. We present the simulation results of three studies and the computation time of more than three studies in Section E.3 and E.4 in S1 Appendix, respectively. We observe that STAREG is feasible for 10 studies or so. We leave more computationally efficient extensions for future research.

## Materials and methods

### Model and algorithm

Suppose we have two datasets obtained by measuring the spatial expression patterns of $m$ genes under distinct cellular environments with different technologies. For each dataset, we have $m$ $p$-values, where the null hypothesis for the $i$th gene states that it is not a spatially variable gene, and the non-null hypothesis states that it is a spatially variable gene. We assume $p$-values from the null follow the standard uniform distribution and $p$-values from the non-null are stochastically smaller than the standard uniform distribution as smaller $p$-values indicate stronger evidence against the null [45]. Denote paired $p$-values as $(p_{1i}, p_{2i})$, $i = 1, \ldots, m$. For $i$th gene, let $\theta_{ji}$ be its hidden state in study $j$ ($j = 1, 2$), where $\theta_{ji} = 1$ indicates $i$th gene is significant in study $j$ and $\theta_{ji} = 0$ otherwise. We assume a two-group model for the two $p$-value sequences, respectively, where

$$p_{1i} \mid \theta_{1i} \sim (1 - \theta_{1i})f_0 + \theta_{1i}f_1,$$
$$p_{2i} \mid \theta_{2i} \sim (1 - \theta_{2i})f_0 + \theta_{2i}f_2, \quad i = 1, \ldots, m,$$

where $f_0$ is the density function of $p$-values under the null, and $f_1$ and $f_2$ denote the non-null density functions for study 1 and study 2, respectively. The two studies share the same $p$-value distribution under the null. The heterogeneity across the two studies is accommodated through modeling density functions under the non-null separately by $f_1$ and $f_2$. Let $\tau_i = (\theta_{1i}, \theta_{2i})$, $i = 1, \ldots, m$ denote the joint hidden states across two studies with prior probabilities $P(\tau_i = (k, l)) = \xi_{kl}$, where $k, l = 0, 1$ and $\sum_{k,l} \xi_{kl} = 1$, such that $\tau_i \in \{(0, 0), (0, 1), (1, 0), (1, 1)\}$. The replicability null hypothesis is

$$H_{i0} : \tau_i \in \{(0, 0), (0, 1), (1, 0)\}, i = 1, \ldots, m.$$

Lfdr is defined as the posterior probability of being null given data. Let $f^i = \xi_{00}f_0(p_{1i})f_0(p_{2i}) + \xi_{01}f_0(p_{1i})f_2(p_{2i}) + \xi_{10}f_1(p_{1i})f_0(p_{2i}) + \xi_{11}f_1(p_{1i})f_2(p_{2i})$, we have

$$\text{Lfdr}_i(p_{1i}, p_{2i}) := 1 - P(\theta_{1i} = \theta_{2i} = 1 \mid p_{1i}, p_{2i})$$
$$= \frac{\xi_{00}f_0(p_{1i})f_0(p_{2i}) + \xi_{01}f_0(p_{1i})f_2(p_{2i}) + \xi_{10}f_1(p_{1i})f_0(p_{2i})}{f^i}. \quad (1)$$

We assume the monotone likelihood ratio condition [27, 45, 46]:

$$f_1(x)/f_0(x) \text{ and } f_2(x)/f_0(x) \text{ are non-increasing in } x. \quad (2)$$

Let $\preceq$ be the elementwise inequality in $R^2$ ($x \preceq y$ if and only if $x_k \leq y_k$ for $k = 1, 2$). A function $g : R^2 \to R$ is monotone for this partial ordering if $x \preceq y$ implies $g(x) \leq g(y)$. From (2), we have that $\text{Lfdr}_i(p_{1i}, p_{2i})$ is monotonically non-decreasing in $(p_{1i}, p_{2i})$. In specific, we have

$$\text{Lfdr}(x_1, x_2) = 1 - \frac{\xi_{11}f_1(x_1)f_2(x_2)}{f(x_1, x_2)},$$

where $f(x_1, x_2) = \xi_{00}f_0(x_1)f_0(x_2) + \xi_{01}f_0(x_1)f_2(x_2) + \xi_{10}f_1(x_1)f_0(x_2) + \xi_{11}f_1(x_1)f_2(x_2)$. If $x_1 \leq y_1$ and

$x_2 \leq y_2$, we aim to show that $\text{Lfdr}(x_1, x_2) \leq \text{Lfdr}(y_1, y_2)$. To this end, we have

$$
\begin{aligned}
&\text{Lfdr}(y_1, y_2) - \text{Lfdr}(x_1, x_2) \\
=\ & \frac{\xi_{11}\{f_1(x_1)/f_0(x_1)\}\{f_2(x_2)/f_0(x_2)\}}{\xi_{00} + \xi_{01}f_2(x_2)/f_0(x_2) + \xi_{10}f_1(x_1)/f_0(x_1) + \xi_{11}\{f_1(x_1)/f_0(x_1)\}\{f_2(x_2)/f_0(x_0)\}} \\
&- \frac{\xi_{11}\{f_1(y_1)/f_0(y_1)\}\{f_2(y_2)/f_0(y_2)\}}{\xi_{00} + \xi_{01}f_2(y_2)/f_0(y_2) + \xi_{10}f_1(y_1)/f_0(y_1) + \xi_{11}\{f_1(y_1)/f_0(y_1)\}\{f_2(y_2)/f_0(y_0)\}}. \\
\geq\ & 0
\end{aligned}
$$

under assumption (2).

The rejection rule based on $\text{Lfdr}_i$ to test replicability null $H_{i0}$ is

$$
\delta_i = I\{\text{Lfdr}_i \leq \lambda\}, \tag{3}
$$

where $\lambda$ is a threshold to be determined. We write the total number of rejections as $R(\lambda) = \sum_{i=1}^{m} I\{\text{Lfdr}_i \leq \lambda\}$, and the number of false rejections as $V(\lambda) = \sum_{i=1}^{m} I\{\text{Lfdr}_i \leq \lambda\}(1 - \theta_{1i}\theta_{2i})$. We have

$$
\begin{aligned}
& E\left[\sum_{i=1}^{m} I\{\text{Lfdr}_i \leq \lambda\}(1 - \theta_{1i}\theta_{2i})\right] \\
=\ & \sum_{i=1}^{m} P(\text{Lfdr}_i \leq \lambda, \theta_{1i}\theta_{2i} = 0) \\
=\ & \sum_{i=1}^{m} \Big\{ \xi_{00}P(\text{Lfdr}_i \leq \lambda \mid \theta_{1i} = 0, \theta_{2i} = 0) + \xi_{01}P(\text{Lfdr}_i \leq \lambda \mid \theta_{1i} = 0, \theta_{2i} = 1) \\
& + \xi_{10}P(\text{Lfdr}_i \leq \lambda \mid \theta_{1i} = 1, \theta_{2i} = 0) \Big\} \\
=\ & \sum_{i=1}^{m} \Big\{ \xi_{00}E[I\{\text{Lfdr}_i \leq \lambda \mid \theta_{1i} = 0, \theta_{2i} = 0\}] + \xi_{01}E[I\{\text{Lfdr}_i \leq \lambda \mid \theta_{1i} = 0, \theta_{2i} = 1\}] \\
& + \xi_{10}E[I\{\text{Lfdr}_i \leq \lambda \mid \theta_{1i} = 1, \theta_{2i} = 0\}] \Big\} \\
=\ & \sum_{i=1}^{m} \Big\{ \int \xi_{00}I\{\text{Lfdr}_i \leq \lambda\}f_0(p_{1i})f_0(p_{2i})dp_{1i}dp_{2i} + \int \xi_{01}I\{\text{Lfdr}_i \leq \lambda\}f_0(p_{1i})f_2(p_{2i})dp_{1i}dp_{2i} \\
& + \int \xi_{10}I\{\text{Lfdr}_i \leq \lambda\}f_1(p_{1i})f_0(p_{2i})dp_{1i}dp_{2i} \Big\} \\
=\ & \sum_{i=1}^{m} \int [\xi_{00}f_0(p_{1i})f_0(p_{2i}) + \xi_{01}f_0(p_{1i})f_2(p_{2i}) + \xi_{10}f_1(p_{1i})f_0(p_{2i})]I\{\text{Lfdr}_i \leq \lambda\}dp_{1i}dp_{2i} \\
=\ & \sum_{i=1}^{m} \int f(p_{1i}, p_{2i})\text{Lfdr}_i I\{\text{Lfdr}_i \leq \lambda\}dp_{1i}dp_{2i} \\
=\ & E\left[\sum_{i=1}^{m} \text{Lfdr}_i I\{\text{Lfdr}_i \leq \lambda\}\right].
\end{aligned}
$$

Write $a \vee b = \max\{a, b\}$. To control FDR of the replicability analysis, we need to find the

critical value $\lambda$ in (3). We estimate FDR by

$$\text{FDR}^*(\lambda) = \frac{\sum_{i=1}^m \text{Lfdr}_i I\{\text{Lfdr}_i \leq \lambda\}}{\sum_{i=1}^m I\{\text{Lfdr}_i \leq \lambda\}},$$

and define $\lambda_m = \sup\{\lambda \in [0, 1] : \text{FDR}^*(\lambda) \leq \alpha\}$. Reject $H_{i0}$ if $\text{Lfdr}_i \leq \lambda_m$. This is the oracle case that assumes we know $\xi_{00}, \xi_{01}, \xi_{10}, \xi_{11}, f_1$ and $f_2$. We provide estimates of them in next section.

## Estimates of unknowns and an adaptive procedure

Assume $f_0$ follows a standard uniform distribution. Let $\boldsymbol{p}_1 = \{p_{1i}\}_{i=1}^m$ and $\boldsymbol{p}_2 = \{p_{2i}\}_{i=1}^m$ denote $p$-values from study 1 and study 2, respectively. Denote hidden states $\boldsymbol{\theta}_1 = \{\theta_{1i}\}_{i=1}^m$ and $\boldsymbol{\theta}_2 = \{\theta_{2i}\}_{i=1}^m$. Under conditional independence of two $p$-value sequences given hidden states, the joint log likelihood function of $(\boldsymbol{p}_1, \boldsymbol{p}_2, \boldsymbol{\theta}_1, \boldsymbol{\theta}_2)$ is

$$
\begin{aligned}
&l(\boldsymbol{p}_1, \boldsymbol{p}_2, \boldsymbol{\theta}_1, \boldsymbol{\theta}_2) \\
=\ & \sum_{i=1}^m [\log\{(1 - \theta_{1i})f_0(p_{1i}) + \theta_{1i}f_1(p_{1i})\} \\
&+ \log\{(1 - \theta_{2i})f_0(p_{2i}) + \theta_{2i}f_2(p_{2i})\} + \theta_{1i}(1 - \theta_{2i}) \log \xi_{10} \\
&+ (1 - \theta_{1i})\theta_{2i} \log \xi_{01} + (1 - \theta_{1i})(1 - \theta_{2i}) \log \xi_{00} + \theta_{1i}\theta_{2i} \log \xi_{11}],
\end{aligned}
$$

where hidden states $\boldsymbol{\theta}_1$ and $\boldsymbol{\theta}_2$ are latent variables. For scalable computation, we utilize EM algorithm [28] in combination of pool-adjacent-violator-algorithm (PAVA) to efficiently estimate the unknowns $(\xi_{00}, \xi_{01}, \xi_{10}, \xi_{11}, f_1, f_2)$ incorporating the monotone likelihood assumption (2) for $f_1$ and $f_2$ (see Section A in S1 Appendix for details). $f_1$ and $f_2$ are estimated non-parametrically, which provides more flexibility. With the estimates $(\widehat{\xi}_{00}, \widehat{\xi}_{01}, \widehat{\xi}_{10}, \widehat{\xi}_{11}, \widehat{f}_1, \widehat{f}_2)$, we obtain Lfdr estimates by:

$$\widehat{\text{Lfdr}}_i = \frac{\widehat{\xi}_{00}f_0(p_{1i})f_0(p_{2i}) + \widehat{\xi}_{01}f_0(p_{1i})\widehat{f}_2(p_{2i}) + \widehat{\xi}_{10}\widehat{f}_1(p_{1i})f_0(p_{2i})}{\widehat{f}},$$

where $\widehat{f} = \widehat{\xi}_{00}f_0(p_{1i})f_0(p_{2i}) + \widehat{\xi}_{01}f_0(p_{1i})\widehat{f}_2(p_{2i}) + \widehat{\xi}_{10}\widehat{f}_1(p_{1i})f_0(p_{2i}) + \widehat{\xi}_{11}\widehat{f}_1(p_{1i})\widehat{f}_2(p_{2i})$. An estimate of $\lambda_m$ can be obtained through

$$\widehat{\lambda}_m = \sup\left\{\lambda \in [0, 1] : \frac{\sum_{i=1}^m \widehat{\text{Lfdr}}_i I\{\widehat{\text{Lfdr}}_i \leq \lambda\}}{\sum_{i=1}^m I\{\widehat{\text{Lfdr}}_i \leq \lambda\}} \leq \alpha\right\}.$$

The replicability null hypothesis $H_{i0}$ is rejected if $\widehat{\text{Lfdr}}_i \leq \widehat{\lambda}_m$. This is equivalent to the step-up procedure [27]: let $\widehat{\text{Lfdr}}_{(1)} \leq \ldots \leq \widehat{\text{Lfdr}}_{(m)}$ be the ordered statistics of $\{\widehat{\text{Lfdr}}_i\}_{i=1}^m$ and denote by $H_{(1)}, \ldots, H_{(m)}$ the corresponding ordered replicability null hypotheses, the procedure works as follows.

$$\text{Find} \quad \widehat{k} := \max\left\{1 \leq i \leq m : \frac{1}{i}\sum_{j=1}^i \widehat{\text{Lfdr}}_{(j)} \leq \alpha\right\},$$

$$\text{reject } H_{(i)} \text{ for } i = 1, \ldots, \widehat{k}.$$

## Supporting information

**S1 Fig. Plots of non-null density functions estimated by STAREG for two pairs of SRT data measured with different technologies.** In each panel, the x-axis shows the $\log_{10}$ scale of $p$-values, and the y-axis shows the $10^{12}$ scale of non-null densities.
(EPS)

**S2 Fig. Additional validation results of the ST data from MOB.** (**A**) Three spatial patterns summarized based on the 163 replicable SVGs uniquely identified by STAREG (left: ST Replicate 1 study; right: ST Replicate 8 study). (**B**) Spatial expression patterns of 16 genes randomly selected from the 163 replicable SVGs uniquely identified by STAREG based on the ST Replicate 1 data. Different color represents relative gene expression levels (antique white: low; navy blue: high). (**C**) Spatial expression patterns of 16 genes randomly selected from the 163 replicable SVGs uniquely identified by STAREG based on the ST Replicate 8 data. (**D**) The percentage of validated genes based on top-2, 000 gene set calculated from the maximum of $p$-values and Lfdr in the MOB data analysis. (**E**) GO terms enriched by the 163 replicable SVGs only identified by STAREG. The horizontal dashed line represents an FDR cutoff of 0.05.
(EPS)

**S3 Fig. Additional validation results of the Slide-seq and Slide-seqV2 data from mouse cerebellum.** (**A**) Three spatial patterns summarized based on the 341 replicable SVGs uniquely identified by STAREG (left: Slide-seq study; right: Slide-seqV2 study). (**B**) Spatial expression patterns of 10 genes randomly selected from the 341 replicable SVGs uniquely identified by STAREG based on the Slide-seq data. Different color represents relative gene expression levels (antique white: low; navy blue: high). (**C**) Spatial expression patterns of 10 genes randomly selected from the 341 replicable SVGs uniquely identified by STAREG based on the Slide-seqV2 data. (D) The percentage of validated genes based on top-2, 000 gene set calculated from the maximum of $p$-values and Lfdr in the MOB data analysis. The validations were based on the reference gene lists from [43] and [44]. (**E**) GO terms enriched by the 341 replicable SVGs only identified by STAREG. The horizontal dashed line represents an FDR cutoff of 0.05.
(EPS)

**S1 Appendix. Supplementary materials.** Including the detailed derivations of STAREG and competing methods, additional simulation studies, and data analysis results.
(PDF)

## Author Contributions

**Conceptualization:** Yan Li, Xianyang Zhang, Hongyuan Cao.

**Data curation:** Yan Li.

**Formal analysis:** Yan Li.

**Funding acquisition:** Hongyuan Cao.

**Investigation:** Hongyuan Cao.

**Methodology:** Yan Li, Hongyuan Cao.

**Project administration:** Hongyuan Cao.

**Resources:** Hongyuan Cao.

**Software:** Yan Li, Hongyuan Cao.

**Supervision:** Xiang Zhou, Rui Chen, Xianyang Zhang, Hongyuan Cao.

**Validation:** Yan Li.

**Visualization:** Yan Li.

**Writing – original draft:** Yan Li, Hongyuan Cao.

**Writing – review & editing:** Yan Li, Xiang Zhou, Rui Chen, Xianyang Zhang, Hongyuan Cao.

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
