## [Decision Letter · Decision Letter 0]

21 May 2024

Dear Dr Cao,

Thank you very much for submitting your Research Article entitled 'Statistical replicability analysis of high throughput experiments with applications to spatial transcriptomic studies' to PLOS Genetics.

The manuscript was fully evaluated at the editorial level and by independent peer reviewers. The reviewers appreciated the attention to an important problem, but raised some substantial concerns about the current manuscript. Based on the reviews, we will not be able to accept this version of the manuscript, but we would be willing to review a much-revised version. We cannot, of course, promise publication at that time.

Should you decide to revise the manuscript for further consideration here, your revisions should address the specific points made by each reviewer. We will also require a detailed list of your responses to the review comments and a description of the changes you have made in the manuscript, focusing particularly on concerns raised by both Reviewer 2 and Reviewer 3 regarding the simplicity of the simulation studies. Provide additional analysis to thoroughly address this issue. Additionally, Reviewer 2 and 3 have expressed concerns about the applicability of the framework to more than two studies/datasets. Please respond to their comments by discussing how the method can be extended to a more general scenario.

If you decide to revise the manuscript for further consideration at PLOS Genetics, please aim to resubmit within the next 60 days, unless it will take extra time to address the concerns of the reviewers, in which case we would appreciate an expected resubmission date by email to plosgenetics@plos.org.

We are sorry that we cannot be more positive about your manuscript at this stage. Please do not hesitate to contact us if you have any concerns or questions.

Yours sincerely,

Jian Hu

Guest Editor

PLOS Genetics

Xiaofeng Zhu

Section Editor

PLOS Genetics

Reviewer's Responses to Questions

**Comments to the Authors:**

Reviewer #1: In this paper, the authors present a new method for detecting replicable signals across two related studies, and illustrate its application to replicable SVG detection. The method is empirical Bayes-based, aims to jointly analyze a pair of p-values. The method seems to be statistical rigorous and computational scalable. Detect replicable biological findings is of interest and method definitely has merits. However, I have the following questions.

I found it rather puzzling that the authors seem to name their method STAREG in the Introduction section and mentioned a few times. But this name completely disappeared in Results, Methods and discussion sections. In all the figures, the authors used “our method” as the label. This is very strange and confusing. Please be consistent.

Lines 102-103, “Suppose we have two datasets obtained by measuring the spatial expression patterns of m genes under distinct cellular environments with different technologies”. This is problematic. By distinct, are authors referring to two different tissues, or under different conditions? If so, then it is expected that a gene may be SVG in one experiment, but non-SVG is the other. Since it is highly likely that a gene being SVG is context-dependent. I would be more comfortable if the two experiments are highly-related.

Figure 2. It seems that line style changed from a, b to c? Why not consistent?

Lines 169-174. I was very surprised to see that the numbers of genes and spots do not change at all after filtering. Are you sure? If so, what is the point of filtering?

Line 177-179. I wonder how many genes identified by other methods as SVG, but not by STAREG?

Reviewer #2: Li et al. introduces a statistical method for identifying replicable differential genes. While the method aims addresses the issue of replicability across studies and demonstrates solidity, the rationale for focusing on spatial transcriptomics data lacks clear justification. Moreover, additional real data applications are in need to strengthen the manuscript's support for the method's accuracy.

I wonder why the authors choose spatial transcriptomics as the application to demonstrate the performance of this method. Even when a gene is spatially variable in two studies, its spatial expression patterns can be quite different, so such a test may not be of strong interest in real practice. A more useful application would be single-cell gene expression analysis, where one tries to determine is a gene is up- or down- regulated in both studies.

The current method considers the scenario with two studies. However, many joint single-cell analyses involve more than two studies/datasets. How easy is it to extend the method for a more general scenario?

The manuscript focuses on single-cell analysis as the application, but the simulation is based on simple normal distributions. There have been many single-cell data simulators available. What if the authors used some of these tools to generate data that mimic real single-cell data characteristics?

An important assumption of the proposed method is that the p values are uniformly distributed under the null. We know that this is often more or less violated when a method is applied to real data. Therefore, in addition to SPARK, it would be helpful to apply additional SVG methods to real data, and evaluate how the proposed methods perform given p values produced by different methods. How much would the results differ between each other?

In both real data applications, it would be interesting to see more detailed analysis on the replicable SVGs that were only identified by the proposed method. Do they show strong spatial patterns? Do they show consistent patterns in both datasets?

Minor:

The authors should provide a more detailed proof or explanation in a supplementary file. For example, for the monotonicity of Lfdr(p1, p2), the formula below formula (3) (page 19), and the estimation of f1 and f2.

Reviewer #3: Uploaded as an attachment.

**Have all data underlying the figures and results presented in the manuscript been provided?**

Reviewer #1: Yes

Reviewer #2: None

Reviewer #3: Yes

PLOS authors have the option to publish the peer review history of their article (what does this mean?). If published, this will include your full peer review and any attached files.

Reviewer #1: No

Reviewer #2: No

Reviewer #3: No

---

## [Decision Letter · Decision Letter 1]

4 Sep 2024

Dear Dr Cao,

Thank you very much for submitting your Research Article entitled 'STAREG: statistical replicability analysis of high throughput experiments with applications to spatial transcriptomic studies' to PLOS Genetics.

The manuscript was fully evaluated at the editorial level and by independent peer reviewers. The reviewers are generally satisfied with the responses. Reviewer 2  has a minor suggestion that we hope you can address before we formally accept the manuscript.

We therefore ask you to modify the manuscript according to the review recommendations. Your revisions should address the specific points made by each reviewer.

To resubmit, log into your Editorial Manager account and select the option 'Revise Submission' in the 'Submissions Needing Revision' folder.

Yours sincerely,

Jian Hu

Guest Editor

PLOS Genetics

Xiaofeng Zhu

Section Editor

PLOS Genetics

We have not yet received feedback from Reviewer 3. Therefore, Reviewer 1 and the editor have reviewed the response and agree that the comments from Reviewer 3 are adequately addressed. Please address the remaining comments from Reviewer 2 in the next revision.

Reviewer's Responses to Questions

**Comments to the Authors:**

Reviewer #1: The authors have addressed all my comments.

Reviewer #2: I appreciate the authors’ response to my previous comments and they have addressed most of my concerns. I just have one additional question. The revised manuscript presents a new analysis to identify marker genes from two mouse aortic leukocyte samples. The analysis reveals that STAREG identified 162 genes that were not detected by any other method in the comparison. Can the authors demonstrate these 162 genes were indeed more highly expressed in T memory cells and biologically relevant?

**Have all data underlying the figures and results presented in the manuscript been provided?**

Reviewer #1: Yes

Reviewer #2: None

PLOS authors have the option to publish the peer review history of their article (what does this mean?). If published, this will include your full peer review and any attached files.

Reviewer #1: No

Reviewer #2: No

---

## [Editor Report · Decision Letter 2]

10 Sep 2024

Dear Dr Cao,

We are pleased to inform you that your manuscript entitled "STAREG: statistical replicability analysis of high throughput experiments with applications to spatial transcriptomic studies" has been editorially accepted for publication in PLOS Genetics. Congratulations!

Yours sincerely,

Jian Hu

Guest Editor

PLOS Genetics

Xiaofeng Zhu

Section Editor

PLOS Genetics

Comments from the reviewers (if applicable):

**Data Deposition**

http://datadryad.org/submit?journalID=pgenetics&manu=PGENETICS-D-24-00265R2

**Press Queries**

---

## [Editor Report · Acceptance letter]

24 Sep 2024

PGENETICS-D-24-00265R2 

STAREG: statistical replicability analysis of high throughput experiments with applications to spatial transcriptomic studies 

Dear Dr Cao, 

We are pleased to inform you that your manuscript entitled "STAREG: statistical replicability analysis of high throughput experiments with applications to spatial transcriptomic studies" has been formally accepted for publication in PLOS Genetics! Your manuscript is now with our production department and you will be notified of the publication date in due course.

With kind regards,

Anita Estes

PLOS Genetics

On behalf of:
